# Robust Link Prediction over Noisy Hyper-Relational Knowledge Graphs

## ABSTRACT

Modern Knowledge Graphs (KGs) are inevitably noisy due to the nature of their construction process. Such noise could significantly impair the performance of link prediction over KGs. Existing robust learning techniques for noisy KGs mostly focus on triple facts, where the fact-wise confidence is straightforward to evaluate. However, hyper-relational facts, where an arbitrary number of key-value pairs are associated with a base triplet, have become increasingly popular in modern KGs, but significantly complicate the confidence assessment of the fact. Against this background, we study the problem of robust link prediction over noisy hyper-relational KGs, and propose NYLON, a Noise-resistant hYper-reLatiONal link prediction technique via active crowd learning. Specifically, beyond the traditional fact-wise confidence, we first introduce element-wise confidence measuring the fine-grained confidence of each entity or relation of a hyper-relational fact. We connect the element- and fact-wise confidences via a "least confidence" principle to allow efficient crowd labeling. NYLON is then designed to systematically integrate three key components, where a hyper-relational link predictor uses the fact-wise confidence for robust prediction, a cross-grained confidence evaluator predicts both element- and fact-wise confidences, and an effort-efficient active labeler selects informative facts for crowd annotators to label using an efficient labeling mechanism guided by the element-wise confidence under the "least confidence" principle and further followed by data augmentation. We evaluate NYLON on three real-world KG datasets against a sizeable collection of baselines. Results show that NYLON achieves superior and robust performance in both link prediction and error detection tasks on noisy KGs, and outperforms best baselines by 2.42-10.93% and 3.46-10.65% in the two tasks, respectively.

## CCS CONCEPTS

• **Computing methodologies → Knowledge representation and reasoning**.

## KEYWORDS

Hyper-relation, Noisy knowledge graph, Link prediction

**ACM Reference Format:**
. 2018. Robust Link Prediction over Noisy Hyper-Relational Knowledge Graphs. In *Proceedings of Make sure to enter the correct conference title from your rights confirmation email (Conference acronym 'XX)*. ACM, New York, NY, USA, 12 pages. https://doi.org/XXXXXXX.XXXXXXX

## 1 INTRODUCTION

Knowledge Graphs (KGs) have been widely used to build various Web applications ranging from Web search [62] to recommendation systems [68]. Traditionally, KGs are represented as a set of triplets, where each triplet *(head, relation, tail)*, or *(h,r,t)* for short, represents a fact that encodes a relation connecting a head entity to a tail entity, such as *(Apple Inc., headquarter location, Cupertino)*. To better describe the rich information of the complex facts in real-world scenarios, modern KGs often contain hyper-relational facts [11, 20, 21, 32, 41, 60, 70], where a base triplet $(h, r, t)$ is further associated with an arbitrary number of key-value $(k, v)$ pairs describing additional information about the base triplet, represented as $(h, r, t, k_1, v_1, ...)$. For example, a hyper-relational fact on Wikidata[1] *(Apple Inc., industry, software industry, in the scope of, computer program, in the scope of, operating system)* involves a base triplet *(Apple Inc., industry, software industry)*, and key-value pairs[2] *(in the scope of, computer program)* and *(in the scope of, operating system)* further describing the detailed information about the scope of the software industry that Apple Inc. is in. To effectively make use of such KGs, link prediction tasks [2, 55] have been widely adopted to solve KG completion and reasoning problems, such as $(h, r, ?)$ or $(h, ?, t, k_1, v_1, ...)$, where the question mark indicates the missing element (entity or relation) to be predicted. Existing approaches to this problem usually design KG embedding models [55] learning to capture the structural information of the KG for predicting the missing element.

Despite the wide adoption of KGs in various domains, modern KGs often contain inevitable noises, which could significantly impair the performance of the downstream applications. Specifically, modern KGs usually contain millions of entities with billions of facts connecting them; such a large scale makes it infeasible for manual knowledge extraction and curation by human experts. Subsequently, existing KGs are either automatically extracted from large Web corpora using heuristic algorithms such as NELL [9] and YAGO [49], or collectively built in a crowdsourcing manner such as Wikidata [52], where both approaches intrinsically result in noisy KGs. On one hand, the automatic extraction approach suffers from noisy source corpora and imperfect extraction techniques. For example, NELL reports an estimated precision of 74%, corresponding to around 0.6 million noisy triplets [9]; YAGO reports an accuracy of 95%, corresponding to around 7.5 million noisy triplets [52]. On the other hand, the crowdsourcing approach is sensitive to participants' motivation and vulnerable to malicious participants. For example, Wikidata has been vandalized frequently as its facts can be freely edited by anyone [24]. Such noises could be very harmful to downstream applications [66].

To tackle the noise problem in KGs, existing work focuses either on noisy triplet detection (a.k.a. error detection) [69] predicting the erroneous triplets, or on robust link prediction over noisy KGs [61]

---

[1]https://www.wikidata.org/wiki/Q312
[2]Note that in a $(k, v)$ pair, $k$ and $v$ are indeed a relation and an entity, respectively.

predicting the confidence of triplets which are then used to improve the downstream link prediction performance. In both approaches, the core problem of confidence assessment of triplets is usually addressed using KG topology [61] and KG embeddings [23], as well as using additional information beyond the noisy KG itself such as KG schematic rules [26], Web corpora [29], third-party clean KGs [19], KG editing history [36], pre-trained large language models [3], or crowdsourcing [19]. Although these existing works propose robust learning solutions to handle noisy KGs, they all focus on triple facts only, which have been shown to oversimplify the complex nature of the real-world facts [41]. However, different from a triple fact that is easy to check its correctness, a hyper-relational fact is much more complex to be checked. More precisely, it is straightforward to check the correctness of a triplet $(h, r, t)$ by assessing whether the entity $h$ and $t$ should be connected via the relation $r$ (i.e., triple confidence [61]). In contrast, for a hyper-relational fact $(h, r, t, k_1, v_1, ...)$, its correctness depends on the compatibility of all its contained elements. This significantly complicates the confidence assessment of the fact, which hinders the application of many existing confidence assessment techniques. For example, a widely adopted relation-path-based approach [61] shows suboptimal performance on hyper-relational KGs (evidenced by our experiments below), as it fails to consider the hyper-relationality of the facts.

In this context, confidence assessment by humans via crowdsourcing becomes a promising solution. However, due to the expensive cost of crowd annotators, how to effectively integrate crowdsourcing into hyper-relational link prediction techniques while maximally benefiting from the limited human labeling budget is still a non-trivial task. On one hand, as a hyper-relational fact could contain an arbitrary number of key-value pairs associated with the base triplet, crowd annotators may need to spend much effort in checking those facts containing a large number of elements (entities and relations), such as the largest hyper-relational fact in our experiment dataset containing 67 entities and 66 relations. It is thus important to design technical solutions to assist crowd annotators in the checking process. On the other hand, the tremendous number of facts contained in modern KGs often largely exceeds the labeling capacity of crowd annotators, where only a few representative facts could be labeled. Subsequently, it is critical to not only select the most informative samples for confidence assessment, but also design an efficient labeling mechanism boosting the confidence prediction performance on other unlabeled facts, so as to improve the ultimate hyper-relational link prediction performance.

Against this background, we propose NYLON, a N̲oise-resistant hY̲per-reL̲atiON̲al link prediction technique via active crowd learning. Specifically, different from existing confidence assessment techniques that only evaluate the confidence of a whole fact (i.e., *fact-wise confidence*), we further introduce so-called *element-wise confidence*, which measures the fine-grained confidence of each element (entity or relation) of a hyper-relational fact, which can be used to significantly reduce the effort of crowd annotators in the noise labeling process. More precisely, we connect the element- and fact-wise confidences using a "*least confidence*" principle, where the confidence of a fact is determined by the least confidence of all its elements; in other words, if one element in a fact is labeled as incorrect, the whole fact is incorrect. Subsequently, this principle allows that for a noisy fact, the crowd annotators may only need to check

part of its elements until an incorrect element is found. Following this principle, NYLON is designed to efficiently and effectively evaluate the confidence of hyper-relational facts for the ultimate goal of resolving hyper-relational link prediction tasks. It consists of three components. First, a *hyper-relational link predictor* is built on top of self-attention networks with a masked training process, where each learning hyper-relational fact is weighted by its fact-wise confidence. Second, a *cross-grained confidence evaluator* learns from a small set of noise-labeled facts to predict both element- and fact-wise confidences. Third, an *effort-efficient active labeler* iteratively selects informative hyper-relational facts for crowd annotators to label according to the fact-wise confidence. To reduce the labeling effort of crowd annotators, element-wise confidence is used to guide crowd annotators to check the elements of a hyper-relational fact according to the ascending order of their element-wise confidence (where the top ones are most probably incorrect), and terminate the labeling process until one incorrect element is found, following the "least confidence" principle. Moreover, the labeler further augments the human-labeled facts by generating pseudo-labeled facts having the same label ratio, which are together regarded as noise-labeled facts to better train the confidence evaluator. In summary, we make the following key contributions:

- We study the problem of robust link prediction over noisy hyper-relational KGs, which is, to the best of our knowledge, the first work on robust learning over noisy hyper-relational KGs.
- We introduce element-wise confidence beyond the traditional fact-wise confidence for hyper-relational facts, and bridge the gap between them using the "least confidence" principle, which could significantly reduce the labeling effort of crowd annotators.
- We design NYLON, a N̲oise-resistant hY̲per-reL̲atiON̲al link prediction technique via active crowd learning. Following the "least confidence" principle, it integrates a hyper-relational link predictor using the fact-wise confidence for robust prediction, a cross-grained confidence evaluator predicting both element- and fact-wise confidences, and an effort-efficient active labeler selecting informative facts for crowd annotators to label via an efficient labeling mechanism followed by data augmentation.
- We conduct a thorough evaluation of NYLON compared to a sizable collection of baselines on three KG datasets. Results show that NYLON outperforms baselines in both link prediction tasks by 2.42-10.93%, and error detection tasks by 3.46-10.65%. It also achieves the best Pareto frontier when trading off the task performance and crowdsourcing labeling effort.

## 2 RELATED WORK

### 2.1 Link Prediction on Hyper-Relational KGs

Hyper-relational KGs encode rich information with hyper-relational facts, where each fact contains multiple relations and entities [22, 41]. Some existing work adopted an n-ary representation for hyper-relational facts, i.e., a set of key-value (relation-entity) pairs [22, 32, 60, 70]. As a typical example in [20, 22], a hyper-relational fact $(h, r, t)$ with $(k, v)$ is transformed into $\{r_h:h, r_t:t, k:v\}$ by converting the relation $r$ into two keys $r_h$ and $r_t$, associated with head $h$ and tail $t$, respectively. Using such n-ary representations, these techniques learn either the relatedness between entity-relation pairs [20, 22], or relatedness among all entities in a fact [32, 60, 70] for

link prediction. However, recent studies [21, 41] revealed that the base triplets $(h, r, t)$ serve as the fundamental data structure in the KGs and preserve the essential information for link prediction, and suggested learning directly from the hyper-relational facts represented as $(h, r, t, k_1, v_1, ...)$. Following this direction, HINGE [41] and NeuInfer [21] design two different feature extraction pipelines for the base triplets and key-value pairs, respectively; StarE [18], Hy-Transformer [67], GRAN [56], and QUAD [47] design Graph Neural Networks (GNNs) to encode the base triplets together with key-value pairs using transformer networks [51] for link prediction.

However, these existing works all assume that the input KG is clean where all facts are correct and noise-free, which is often an unrealistic assumption for real-world large-scale KGs that are either automatically extracted from Web corpora [9, 49] or collectively built in a crowdsourcing manner [52].

## 2.2 Robust Learning on Noisy KGs

To tackle noisy KGs, robust learning techniques are widely studied, which can be classified into two categories according to whether additional information beyond KG facts is used.

The first category relies on facts of KGs only, evaluating the confidence of facts using the topology/structure/paths of a KG and mostly together with the entity/relation embeddings of the KG. For example, CKRL [61] combines local triple confidence measured by the link prediction score and global path confidence measured by the reliability and semantic closeness of the relation path connecting two entities of a triplet; CKRL has been later extended using convolutional neural networks [65] and transformers [63]; KGTtm [31] estimates the confidence of triplets under a PageRank-like resource allocation mechanism; FEA [39] aggregates the embeddings of semantically relevant paths from a head entity for KG error detection; SUKE [54] combines KG structural and knowledge uncertain information for fact confidence prediction; Neil et al. [33] used link-specific learnable bias for robust learning on noisy KGs; Reform [57] designs an error mitigation technique using GNNs for confidence prediction; GEDet [23] combines graph data augmentation and generative adversarial networks for erroneous entity detection; CAGED [69] uses contrastive learning in KG embedding models for KG error detection by focusing on nontrivial erroneous triplets; IDKG [27] combines entity embedding similarity with relation path confidence to detect noisy facts.

Besides the information from the KG, additional data have also been used to design noise-resistant learning techniques. One of the primary pieces of information in this context is the KG schema [25], which is usually formulated as a set of schematic rules for confidence evaluation or error detection [5, 63, 65, 71]. For example, RUGE [26] proposes a schema-rule-based KG cleaning technique to filter out schematically incorrect triplets to improve the quality of KG embeddings; Cheng et al. [10] proposed a rule-based KG repairing method with graph repairing rules, which are generated by AMIE algorithm [17]; Pellissier et al. [37] proposed to use (in-)completeness meta-information to assess the quality of rules learned from incomplete KGs; DSKRL [45] proposes a triple dissimilarity measure based on entity type hierarchy and relation path information for both KG error detection and completion tasks. Besides KG schema, other data sources have also been used. For

example, CrossVal [59] uses an external KG to validate facts in a target KG via cross-graph representation learning; TKGC [29] jointly performs fact extraction tasks and noisy fact cleaning using open Web data; Bass [38] uses KG edit history and a set of constraints, to automatically correct constraint violations of facts; Arnaout et al. [3] exploited pre-trained large language model probes for KG repairing; Knowledge Vault [13] is a Web-scale probabilistic knowledge base where noisy facts are detected and corrected using prior models built from already-cataloged knowledge.

These existing robust learning methods all focus on triple facts only, which have been shown to oversimplify the complex nature of real-world facts [41]. However, hyper-relational KGs significantly complicate the confidence assessment of hyper-relational facts, which hinders the application of many existing confidence assessment techniques. Therefore, we propose NYLON for noise-resistant hyper-relational link prediction via active crowd learning.

## 2.3 Crowdsourcing for KGs

Crowdsourcing techniques have been widely adopted for KG construction [30, 40, 50] and KG alignment [7, 28, 72]. A few existing works also leverage crowdsourcing for cleaning noisy KGs. For example, Acosta et al.[1] adopted a Find-Fix-Verify mechanism [6] to directly fix incorrect/incomplete object values, data types, and links in KGs; WhoKnows [53] proposes a strategy to generate questionnaires for KG data cleaning; KGClean [19] uses a pre-trained clean KG embedding model combined with crowdsourcing to detect and repair a noisy KG; KAEL [12] integrates crowdsourcing with ensemble learning for KG noise detection. In this context, various active learning sampling strategies, such as uncertainty sampling [43] selecting the most confusing samples to a classifier or Farthest-Traversa [48] selecting the most distant samples to the labeled samples in a latent space, could be integrated with crowdsourcing. However, these prior works have not looked into crowdsourcing for noisy hyper-relational KGs work. Our work fills this gap.

A slighted related line of work can be found on addressing the human side of issues in the crowdsourcing literature, for example, various human biases affected by cognitive [15], cultural and demographic [14, 42] factors. Methods for dealing with these biases [4, 64] are orthogonal to our work, in the sense that they can be considered in our problem but are not the focus of our work. Like prior work in KGs [12, 19], our work focuses on the computational challenges in tackling the noise issue in KGs, through automatic, robust link prediction and computing mechanisms for reducing human annotation effort by NYLON.

## 3 NYLON

In this section, we present NYLON, a noise-resistant hyper-relational link prediction technique via active crowd learning. Specifically, beyond the traditional *fact-wise confidence* that is widely used to evaluate the confidence of a fact as a whole, we introduce *element-wise confidence* measuring the fine-grained confidence of each element (entity or relation) of a hyper-relational fact. We connect the *element-wise confidence* to the *fact-wise confidence* using a "least confidence" principle, which states that *the confidence of a fact is determined by the least confidence of all its elements*. Following this principle, we design NYLON integrating three components as

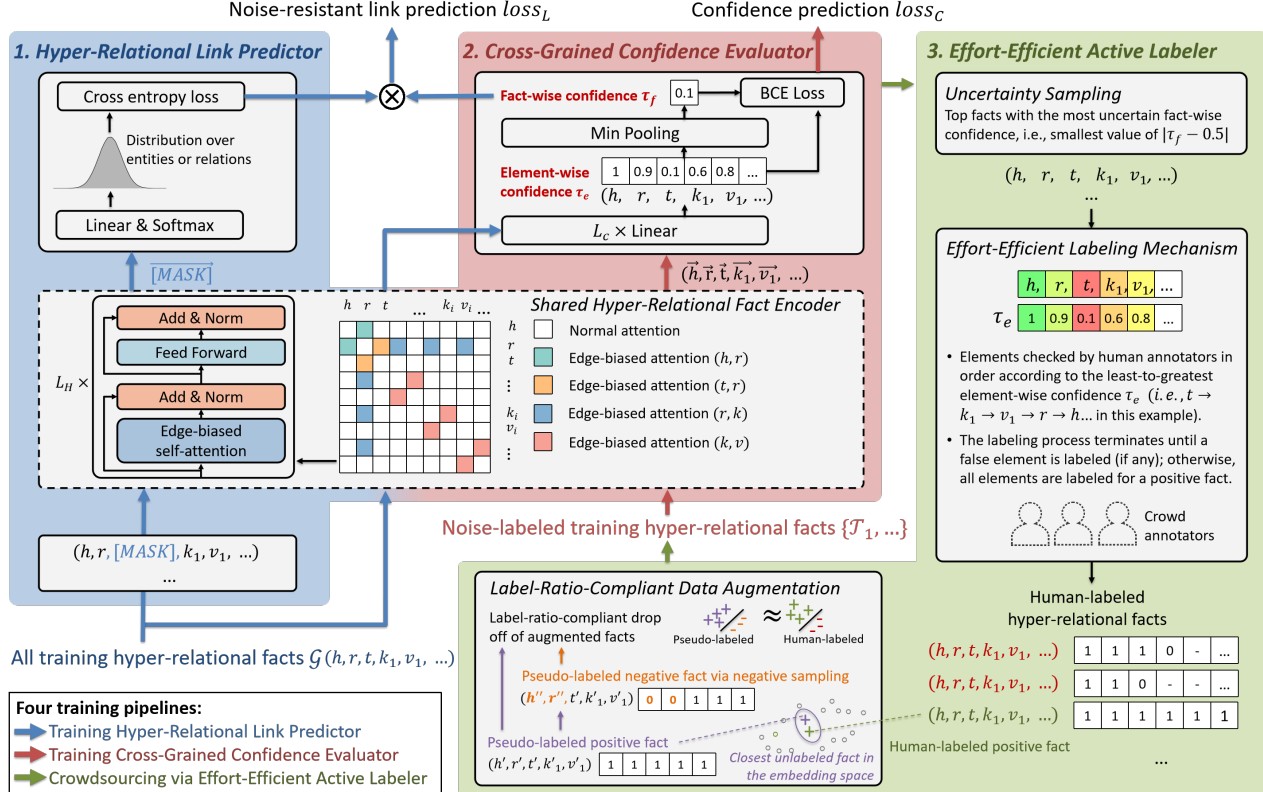

**Figure 1: Overview of NYLON with three components: 1) Hyper-Relational Link Predictor; 2) Cross-Grained Confidence Evaluator; 3) Effort-Efficient Active Labeler. Three training pipelines are shown as arrows in three different colors, respectively.**

shown in Figure 1: 1) a *hyper-relational link predictor* built on top of a hyper-relational fact encoder based on self-attention networks with a masked training process, where each training fact is weighted by its predicted fact-wise confidence; 2) a *cross-grained confidence evaluator* trained on a small set of noise-labeled hyper-relational facts obtained via crowdsourcing, so as to predict both element- and fact-wise confidences while following the "least confidence" principle; 3) an *effort-efficient active labeler* selecting informative hyper-relational facts for crowd annotators to label according to the fact-wise confidence, guiding the labeling process using the element-wise confidence to reduce the labeling effort of crowd annotators, and also augmenting the human-labeled facts by generating pseudo-labeled facts which are together regarded as noise-labeled facts to better train the confidence evaluator. We present the detail of each component below, followed by the model training process.

## 3.1 Hyper-Relational Link Predictor

The hyper-relational link predictor is designed to predict a missing element (entity or relation) in a hyper-relational fact. It is built on top of a hyper-relational fact encoder with a masked training process. Specifically, inspired by [56], our encoder adopts a self-attention network with learnable edge biases discriminating connections between different elements in the hyper-relational fact, so as to capture the correlation between entities and relations both in the base triplet and the key-value pairs. We present a single self-attention layer below. For an input hyper-relational fact

$(h, r, [mask], k_1, v_1, ...)$ (with a missing element masked for prediction), it is fed into a self-attention layer [51]. For each element $u_i \in (h, r, [mask], k_1, v_1, ...)$, its embedding $\vec{u}_i \in \mathbb{R}^d$ is first projected into attention query, key and value[3] $\mathbf{W}^Q\vec{u}_i, \mathbf{W}^K\vec{u}_i, \mathbf{W}^V\vec{u}_i \in \mathbb{R}^d$ by linear transformation parameters $\mathbf{W}^Q, \mathbf{W}^K, \mathbf{W}^V \in \mathbb{R}^{d \times d}$. The pair-wise similarity between elements is computed as:

$$\beta_{ij} = \frac{\left(\mathbf{W}^Q\vec{u}_i\right)^\top \left(\mathbf{W}^K\vec{u}_j + \vec{c}_{ij}^K\right)}{\sqrt{d}} \quad (1)$$

where $\vec{c}_{ij}^K$ (and also $\vec{c}_{ij}^V$ below) refers to learnable edge biases on attention key (and value)[3] [46, 56]. In the self-attention layer, a hyper-relational fact is viewed as a fully-connected graph with edge biases specifying the type of edges connecting different elements in the hyper-relational fact. Five types of undirected edges are considered, namely $(h, r), (t, r), (r, k), (k, v)$ and others not included in the categories above, which is shown as an adjacency matrix in Figure 1. Afterward, a softmax function is used to normalize the similarity score $\beta_{ij}$, and the edge biases $\vec{c}_{ij}^V$ on the attention value[3] is also added when updating the embedding of element $\vec{u}_i$:

$$\vec{u}_i' = \sum_{j=1}^{N} \frac{\exp\left(\beta_{ij}\right)}{\sum_{k=1}^{N} \exp\left(\beta_{ik}\right)} \left(\mathbf{W}^V\vec{u}_j + \vec{c}_{ij}^V\right) \quad (2)$$

---

[3]Note that attention key and value are completely irrelevant to the key-value pairs in a hyper-relational fact.

where $N$ represents the number of elements in the hyper-relational fact. We stack a number of $L_H$ self-attention layers with learnable edge biases to generate the embedding of the [MASK] token, denoted as $\vec{x}_M$. To predict the missing element, a single layer of a linear transformation with a softmax function is used:

$$\vec{p} = \text{softmax}\left(\mathbf{W}_M \vec{x}_M + \vec{b}_M\right) \tag{3}$$

where $\mathbf{W}_M$ is the weight matrix of the input embedding layer of entities and $\vec{b}_M$ is a learnable entity bias, as the missing element is an entity in the above example. For predicting a missing relation, the learnable parameters $\mathbf{W}_M$ and $\vec{b}_M$ in Eq. (3) correspond to the weight matrices of the input embedding layer of relation and relation bias, respectively. The final output of our hyper-relational link predictor is a probability distribution $\vec{p}$ over all entities (or relations), and we compute the cross-entropy loss against the ground truth entity (or relation), denoted as $\vec{y}_l$. To consider the confidence of an input fact, we re-scale its loss according to its predicted fact-wise confidence $\tau_f$ (which we introduce in the next subsection), to form a so-called noise-resistant link prediction $loss_L$ as follows:

$$loss_L = \text{CELoss}(\vec{p}, \vec{y}_l) \cdot \tau_f \tag{4}$$

Subsequently, for a fact with low confidence, our noisy-resistant link prediction loss can weaken its importance in the training process. The detailed training process will be presented in Section 3.4 below.

## 3.2 Cross-Grained Confidence Evaluator

The cross-grained confidence evaluator is designed to predict both element- and fact-wise confidences while following the "least confidence" principle. Due to the expensive cost of crowd annotators, it is trained on a small set of noise-labeled hyper-relational facts obtained via our effort-efficient active labeler. First, the cross-grained confidence evaluator shares the same hyper-relational fact encoder used by the hyper-relational link predictor, due to the following two reasons. On one hand, the encoder generates informative embeddings by learning the complex correlation between elements in a hyper-relational fact, which could then be used to effectively support different downstream tasks including link prediction and confidence prediction, as these two tasks both require learning the correlation and compatibility of elements in the fact. On the other hand, as the small set of noise-labeled hyper-relational facts is often insufficient to train a high-quality encoder for the confidence prediction task, the shared hyper-relational factor encoder could benefit from the large training set for the link prediction task (as evidenced by our ablation study later). Second, based on the embeddings generated by the encoder for all elements in a hyper-relational fact, we adopt a feed-forward network with $L_c$ fully connected layers, noted as $FFN$, to predict the element-wise confidence $\vec{\tau}_e$ of the fact.

$$\vec{\tau}_e = \text{sigmoid}\left(FFN\left([\vec{x}_h, \vec{x}_r, \vec{x}_t \dots]\right)\right) \tag{5}$$

where the sigmoid function bounds the confidence $\vec{\tau}_e \in (0, 1)$. For the fact-wise confidence, following the "least confidence" principle which states that the confidence of a fact is determined by the least confidence of all its elements, we utilize a min-pooling layer to generate the fact-wise confidence $\tau_f$:

$$\tau_f = \min\left(\vec{\tau}_e\right) \tag{6}$$

We train our cross-grained confidence evaluator by defining a loss combining both element-wise confidence $loss_e$ and fact-wise confidence $loss_f$, which are both computed using BCELoss as follows:

$$loss_e = \frac{1}{||\vec{m}_e||_0} \text{BCELoss}\left(\vec{\tau}_e \odot \vec{m}_e, \vec{y}_e \odot \vec{m}_e\right) \tag{7}$$

$$loss_f = \text{BCELoss}(\tau_f, y_f) \tag{8}$$

where $y_f \in \{0, 1\}$ is the fact-wise confidence label, while $\vec{y}_e$ is the element-wise confidence label vector (each entry of $\vec{y}_e$ refers to the confidence label of the corresponding element). Note that $\vec{y}_e$ could be partially labeled, as our "least confidence" principle suggests the labeling process could be terminated until one incorrect element is found, so as to reduce the crowd labeling effort (see Section 3.3 below for more detail). Subsequently, we introduce a binary mask vector $\vec{m}_e$ to discount the impact of unlabeled elements in the loss, where an entry of 1 in $\vec{m}_e$ indicates the corresponding element is actually labeled, 0 otherwise; $\odot$ is the Hadamard product, and $||\vec{m}_e||_0$ is the L0-norm (the number of non-zero values) of $\vec{m}_e$ for normalization. The overall confidence prediction $loss_C$ combines both element- and fact-wise confidence losses as follows:

$$loss_C = loss_e + loss_f \tag{9}$$

## 3.3 Effort-Efficient Active Labeler

The effort-efficient active labeler is designed on one hand for actively selecting a small set of informative hyper-relational facts for crowd annotators to label, while on the other hand for reducing the labeling effort of crowd annotators via an effort-efficient labeling mechanism, followed by label-ratio-compliant data augmentation.

*3.3.1 Uncertainty sampling for active learning.* To select the most informative facts to train the confidence evaluator, we follow the idea of uncertainty sampling [43] for active learning to pick the most uncertain facts. To this end, we select the facts whose fact-wise confidence is most close to 0.5 (which our confidence evaluator is most uncertain of) as follows:

$$f^* = \underset{f \in \mathcal{D}}{\arg\min} \left|\tau_f - 0.5\right| \tag{10}$$

where $\mathcal{D}$ denotes a set of training facts. In practice, for each active learning iteration, we pick the top uncertain and unlabeled facts as query facts for labeling, under a crowdsourcing budget constraint (see Section 3.4 below for more detail).

*3.3.2 Effort-efficient labeling mechanism.* To reduce the labeling effort of crowd annotators, we follow the "least confidence" principle to design an effort-efficient labeling mechanism. Specifically, we use the element-wise confidence of each query fact to guide crowd annotators to check the elements of the fact according to the ascending order of their element-wise confidence (where the top elements are most probably incorrect), and terminate the labeling process until one incorrect element is found, as shown in Figure 1. Subsequently, for a noisy fact, the crowd annotators need to check only part of its elements until an incorrect element is found, thus reducing the labeling effort of crowd annotators. Note that the unlabeled elements do not affect the training process of the cross-grained confidence evaluator, as the unlabeled elements are marked by the binary mask vector $\vec{m}_e$ as 0, which eliminates the loss of unlabeled element from the learning objective using Eq. 7.

*3.3.3 Label-ratio-compliant data augmentation.* Based on the above human-labeled facts, we perform label-ratio-compliant data augmentation, which is particularly beneficial when learning from very limited human-labeled facts, as evidenced by our ablation study later. Specifically, following the widely adopted "cluster assumption" [58, 73] in semi-supervised learning, we first generate $k$ pseudo-labeled positive facts from each human-labeled positive fact, i.e., its top $k$ closest unlabeled facts in the fact embedding space under L2-distance, where fact embeddings are the concatenated output of our the hyper-relational fact encoder $\left[ \vec{x}_h, \vec{x}_r, \vec{x}_t, \ldots \right]$. Afterward, from each pseudo-labeled positive fact, we generate one pseudo-labeled negative fact via negative sampling by randomly corrupting its element. Then, we perform label-ratio-compliant drop-off on these pseudo-labeled facts so as to ensure their positive/negative ratio is consistent with the ratio of human-labeled facts. Finally, the pseudo-labeled facts and the human-labeled facts are together regarded as noise-labeled facts for training the confidence evaluator.

## 3.4 Training Process

*3.4.1 Overall training process.* The training process of NYLON consists of three pipelines as shown in Figure 1. For each training epoch, the three pipelines alternate as follows. First, we start by training our cross-grained confidence evaluator on a small set of noise-labeled training facts (randomly selected in the first epoch, and then generated by our active labeler in the following epochs) under a crowdsourcing budget $b$, to minimize the overall confidence loss integrating both element- and fact-wise confidences using Eq. 9. Second, we train our hyper-relational link predictor using all training facts. Specifically, each training fact is fed both to the hyper-relational link predictor to perform the masked training process obtaining its cross-entropy loss for link prediction, and to the cross-grained confidence evaluator to obtain the element- and fact-wise confidences (which are cached for active labeler later for efficiency purposes). The obtained link prediction loss and the fact-wise confidence are finally combined as the noise-resistant link prediction loss using Eq. 4, which is optimized via backpropagation. Third, our effort-efficient active labeler uses the cached element- and fact-wise confidence to select the top uncertain and unlabeled facts as query facts for human labeling (under the given crowdsourcing budget $b$), which are then augmented to noise-labeled facts. We repeat the training pipelines until convergence.

*3.4.2 Incremental training of confidence evaluator via meta-learning.* In the above training process, the amount of noise-labeled training facts increases over epochs. Here we adopt an incremental training scheme using meta-learning [16, 34] to efficiently train our cross-grained confidence evaluator. Specifically, we follow a similar incremental training scheme for active learning as introduced by [35], where we regard the set of noise-labeled training facts $\mathcal{T}_i$ in the epoch $i$ as one meta-learning task, and set the meta-goal as generalizing the model to the latest $w$ sets of noise-labeled training facts, i.e, $\{\mathcal{T}_l \mid \max(i - w, 1) < l \leq i\}$, where the max operation implies that when the current epoch $i$ is less than the defined window size $w$, we use all existing sets of noise-labeled training facts $\{\mathcal{T}_l \mid 1 < l \leq i\}$. We use the first-order meta-learning algorithm Reptile [34] for efficient parameter updating.

Appendix A.1 summarizes the NYLON training algorithm.

**Table 1: Statistics of the datasets**

| Dataset | JF17K | WikiPeople | WD50K |
|---|---|---|---|
| #Entities / #Relations | 28,645 / 501 | 34,825 / 178 | 47,109 / 531 |
| #Training facts | 76,379 | 294,439 | 166,345 |
| Triple+Hyper (%) | 57.9%+42.1% | 97.4%+2.6% | 86.2%+13.8% |
| #Test tuples | 6,144 | 9,472 | 46,139 |
| Triple+Hyper (%) | 42.4%+57.6% | 97.2%+2.8% | 86.9%+13.1% |

## 4 EXPERIMENTS

### 4.1 Experimental Setup

*4.1.1 Dataset.* We evaluate NYLON on three commonly used hyper-relational KG datasets **JF17K** [70], **WikiPeople** [22] and **WD50K** [18]. The training/validation/test datasets are already split by the data providers. Table 1 shows the dataset statistics.

As there are no explicitly labeled noisy facts in these datasets, we follow a commonly adopted strategy [19, 61, 63] to generate a certain number of noisy facts and insert them into our dataset to form a noisy KG. Specifically, different from the existing noisy fact generation method that randomly corrupts one element (entity or relation) in a positive triple fact to generate a noisy fact, we extend it to a two-step approach for hyper-relational facts. First, for a hyper-relation fact of $n$ elements, we randomly choose $q$ ($q \in \mathbb{N}$ and $1 \leq q \leq \frac{n}{2}$) elements to corrupt, to ensure the number of corrupt elements is not greater than half of the total elements in the fact. Second, we corrupt each chosen element (an entity or a relation) by a randomly picked entity or relation, respectively. Note that for a triple fact where $n = 3$, our approach randomly corrupts $q = 1$ (as $1 \leq q \leq \frac{3}{2}$) element, which is equivalent to existing noisy fact generation methods for triple facts [19, 61, 63]. To evaluate the robustness of our method against different levels of noise, following the setting in [61], we generate and insert different percentages of noisy facts compared to the number of positive facts. Specifically, we consider the cases of noisy facts being 2%, 5%, 10%, 20%, 40%, 60%, 80%, and 100% of the positive facts in our experiments.

*4.1.2 Baselines.* We compare NYLON against a sizeable collection of state-of-the-art techniques from the following three categories. First, *hyper-relational link prediction techniques* without considering noisy facts include **GRAN** [56], **StarE** [18], **Hy-Transformer** [67] and **QUAD** [47]. Second, *robust learning techniques* for both error detection and link prediction over noisy KGs include **CKRL** [61] and our improved version **CKRL-Fix** (by adapting to hyper-relational KGs), **KGTtm** [31], and **IDKG** [27]. Third, *active crowd learning techniques* consider two specific settings for selecting informative data samples to label via crowdsourcing. On one hand, following the setting of the error detection techniques [27, 31], we regard the output of the binary classifier for error detection as the fact-wise confidence which is then used for active learning and crowdsourcing; these methods are denoted as **CKRL-Fix (AL)**, **KGTtm (AL)**, and **IDKG (AL)**. On the other hand, we also consider active learning sampling strategies based on learnt embeddings, Farthest-Traversa (FT) [48] and Density-Weighted Methods (DWM) [44], which are integrated with our NYLON by replacing our uncertainty sampling method, denoted as **NYLON-FT** and **NYLON-DWM**, respectively; we refer to them together with our NYLON as the *NYLON family*. The detailed description and settings of the baselines and our NYLON are presented in the Appendix A.2.

**Table 2: Overall link prediction performance (\*StarE, Hy-Transformer, and QUAD are designed to only predict heads/tails; they cannot be applied for relation prediction (marked as N/A), and their results on entity prediction also exclude value prediction.**

| Method | | JF17K | | | | WikiPeople | | | | WD50K | | | |
|---|---|---|---|---|---|---|---|---|---|---|---|---|---|
| | | Entity | | Relation | | Entity | | Relation | | Entity | | Relation | |
| | | MRR | Hit@1 | MRR | Hit@1 | MRR | Hit@1 | MRR | Hit@1 | MRR | Hit@1 | MRR | Hit@1 |
| Hyper-relational link prediction | GRAN | 0.4910 | 0.4063 | 0.9902 | 0.9851 | 0.4259 | 0.3132 | 0.9490 | 0.9211 | 0.2804 | 0.2171 | 0.8951 | 0.8545 |
| | StarE\* | 0.4322 | 0.3449 | N/A | N/A | 0.3457 | 0.2149 | N/A | N/A | 0.2372 | 0.1669 | N/A | N/A |
| | Hy-Transformer\* | 0.4705 | 0.3848 | N/A | N/A | 0.3825 | 0.2612 | N/A | N/A | 0.2631 | 0.1924 | N/A | N/A |
| | QUAD\* | 0.3869 | 0.2888 | N/A | N/A | 0.3167 | 0.1891 | N/A | N/A | 0.2362 | 0.1693 | N/A | N/A |
| Robust learning | CKRL | 0.4764 | 0.3952 | 0.9896 | 0.9842 | 0.4398 | 0.3502 | 0.9256 | 0.8915 | 0.2726 | 0.2094 | 0.8934 | 0.8528 |
| | CKRL-Fix | 0.4870 | 0.4013 | 0.9905 | 0.9858 | 0.4328 | 0.3208 | 0.9480 | 0.9210 | 0.2815 | 0.2177 | 0.8969 | 0.8561 |
| | KGTtm | 0.4744 | 0.3921 | 0.9877 | 0.9820 | 0.4153 | 0.3012 | 0.9487 | 0.9217 | 0.2654 | 0.2047 | 0.8860 | 0.8431 |
| | IDKG | 0.4875 | 0.4045 | 0.9893 | 0.9840 | 0.4570 | 0.3685 | 0.9454 | 0.9186 | 0.2732 | 0.2167 | 0.8796 | 0.8353 |
| Active crowd learning | CKRL-Fix (AL) | 0.4839 | 0.4048 | 0.9871 | 0.9802 | 0.3832 | 0.2900 | 0.9333 | 0.9045 | 0.2750 | 0.2178 | 0.8771 | 0.8331 |
| | KGTtm (AL) | 0.4776 | 0.3991 | 0.9851 | 0.9773 | 0.3788 | 0.2846 | 0.9350 | 0.9053 | 0.2716 | 0.2150 | 0.8753 | 0.8302 |
| | IDKG (AL) | 0.4894 | 0.4044 | 0.9899 | 0.9849 | 0.4258 | 0.3139 | 0.9470 | 0.9179 | 0.2799 | 0.2164 | 0.8940 | 0.8533 |
| NYLON family | NYLON-FT | 0.5122 | 0.4265 | 0.9917 | 0.9877 | 0.4732 | 0.3795 | 0.9457 | 0.9234 | 0.3064 | 0.2454 | 0.9014 | 0.8641 |
| | NYLON-DWM | 0.5082 | 0.4253 | 0.9905 | 0.9862 | 0.4721 | 0.3847 | 0.9452 | 0.9236 | 0.3035 | 0.2422 | 0.8938 | 0.8561 |
| | NYLON | **0.5349** | **0.4476** | **0.9929** | **0.9894** | **0.5019** | **0.4151** | **0.9607** | **0.9436** | **0.3285** | **0.2649** | **0.9162** | **0.8838** |

*4.1.3 Evaluation Protocol.* We consider hyper-relational link prediction as our primary evaluation task. It predicts a missing element in a hyper-relational fact, such as $(h, r, ?, k_1, v_1, ...)$ or $(h, ?, t, k_1, v_1, ...)$ where the missing element is an entity or a relation, respectively. We generate a ranking list of entities or relations using each method and report two commonly used metrics, i.e., Mean Reciprocal Rank (**MRR**) and **Hits@1**, for link prediction tasks on entities and relations separately. Moreover, we also evaluate confidence prediction as a binary classification task, which is also known as error detection tasks [31], classifying whether a fact is a noisy fact; we report **Accuracy** as the evaluation metric. In addition, as our method NYLON can also predict element-wise confidence beyond the fact-wise confidence, we thus report element-wise accuracy when applicable.

To evaluate the efficiency of our proposed labeling mechanism, we consider different crowdsourcing labeling budgets $b$. Specifically, following our element-wise confidence, we define the $b$ as the number of elements to be labeled in each active learning iteration. To discount the impact of the different numbers of facts across datasets, we define $b$ as a percentage of elements (over the number of all elements of all facts in a dataset) to be labeled and consider the following values: 0.025%, 0.05%, 0.1%, 0.15%, 0.2%, and 0.25%. The default budget is set to 0.25% if not specified otherwise.

## 4.2 Link Prediction Performance

We compare NYLON with all baselines on hyper-relational link prediction performance. Table 2 shows the results with the dataset setting of 100% noise level, which is the highest noisy level and thus the most difficult dataset setting (results with 40% noise level are also shown in Appendix A.3). Figure 2 further shows the performance of the best-performing techniques (per category of techniques on each dataset) across different noise levels.

We observe that NYLON consistently achieves the best link prediction performance compared to all baselines in Table 2. Specifically, NYLON outperforms the best-performing baselines (from all categories except the NYLON family) by 4.93%, 6.52%, and 10.93% on JF17K, WikiPeople, and WD50K, respectively. Moreover, we also observe in Figure 2 that NYLON is much more robust than

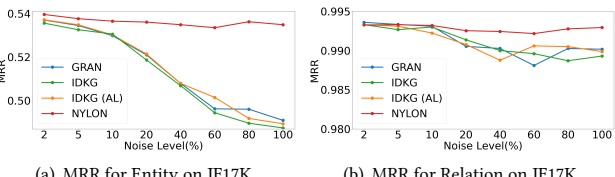

(a) MRR for Entity on JF17K  (b) MRR for Relation on JF17K

**Figure 2: Performance comparison with the best-performing baselines (per category of techniques) on JF17K (results on other datasets shown in Appendix A.4).**

baseline techniques against different levels of noise. When increasing the level of noise, the performance of NYLON decreases much slower than baseline techniques or even retains its performance in some cases. Finally, compared to the best baseline in NYLON family NYLON-FT, NYLON yields 2.42%, 4.81%, and 4.77% improvement (on average) on the three respective datasets, showing the advantage of our uncertainty sampling technique based on the confidence predicted by our cross-grained confidence evaluator.

Interestingly, we observe that the existing robust learning techniques designed for noisy triple facts sometimes yield little improvement on noisy hyper-relational KGs. For example, in Table 2, compared to the best hyper-relational link prediction method GRAN, the best robust learning method IDKG shows comparable results on JF17K and WD50K, but a clear improvement on WikiPeople. Because triple facts dominate the WikiPeople dataset (over 97% as shown in Table 1), which is not the case in the other two datasets.

## 4.3 Error Detection Performance

We evaluate NYLON on the error detection task, which applies to all active crowd learning techniques learning a binary classifier to predict the probability of a fact being fake. Table 3 shows the results with 100% noise level (results with 40% noise level are shown in Appendix A.3). We observe that NYLON significantly outperforms all baselines in this task, yielding 10.65% and 3.46% improvement (on average over datasets) on fact- and element-wise error detection, respectively, over the best baselines. We also find that the methods using hand-crafted features/confidences for triple

**Table 3: Error detection performance (in accuracy)**

| Method | JF17K | | WikiPeople | | WD50K | |
|---|---|---|---|---|---|---|
| | Fact | Element | Fact | Element | Fact | Element |
| CKRL-Fix (AL) | 0.5022 | N/A | 0.5029 | N/A | 0.5014 | N/A |
| KGTtm (AL) | 0.5172 | N/A | 0.6409 | N/A | 0.5308 | N/A |
| IDKG (AL) | 0.7571 | N/A | 0.7965 | N/A | 0.7806 | N/A |
| NYLON-FT | 0.8104 | 0.9166 | 0.8722 | 0.9337 | 0.8511 | 0.9358 |
| NYLON-DWM | 0.8407 | 0.9332 | 0.8565 | 0.9265 | 0.7102 | 0.8568 |
| NYLON | **0.9606** | **0.9791** | **0.9528** | **0.9661** | **0.9231** | **0.9543** |

**Table 4: Ablation study on link prediction (in MRR)**

| Method | JF17K | | WikiPeople | | WD50K | |
|---|---|---|---|---|---|---|
| | Entity | Relation | Entity | Relation | Entity | Relation |
| NYLON (noSE) | 0.5194 | 0.9906 | 0.4825 | 0.9485 | 0.3122 | 0.9004 |
| NYLON (noUS) | 0.5224 | 0.9917 | 0.4908 | 0.9591 | 0.3160 | 0.9061 |
| NYLON (noEEL) | 0.5299 | 0.9926 | 0.5018 | 0.9587 | 0.3238 | 0.9112 |
| NYLON | **0.5349** | **0.9929** | **0.5019** | **0.9607** | **0.3285** | **0.9162** |

**Table 5: Ablation study on error detection (in accuracy)**

| Method | JF17K | | WikiPeople | | WD50K | |
|---|---|---|---|---|---|---|
| | Fact | Element | Fact | Element | Fact | Element |
| NYLON (noSE) | 0.8556 | 0.9307 | 0.9078 | 0.9455 | 0.8185 | 0.9158 |
| NYLON (noUS) | 0.9283 | 0.9700 | 0.9284 | 0.9596 | 0.8861 | 0.9451 |
| NYLON (noEEL) | 0.9480 | 0.9776 | 0.9390 | 0.9651 | 0.9070 | 0.9528 |
| NYLON | **0.9606** | **0.9791** | **0.9528** | **0.9661** | **0.9231** | **0.9543** |

facts, i.e., CKRL-Fix (AL), KGTtm (AL), and IDKG (AL), show much worse results than the NYLON family. Because these heuristic features/confidences for triple facts, such as the widely used relation path reliability, fail to consider the hyper-relationality of our facts. On the WikiPeople dataset dominated by triple facts, these methods indeed achieve a slightly higher accuracy in general, which, however, is still much lower than our NYLON.

## 4.4 Ablation Study

We consider the following variation of NYLON in the ablation study. **NYLON (noSE)** does not use the Share hyper-relational fact Encoder between the hyper-relational link predictor and the cross-grained confidence evaluator; instead, it uses two separate encoders for the two respective components. **NYLON (noUS)** does not use the Uncertainty Sampling technique; instead, it randomly samples facts to be labeled. **NYLON (noEEL)** removes the Effort-Efficient Labeling mechanism, where the element-wise confidence is not used and crowd annotators are required to label all elements for each query fact. Table 4 and 5 show the results on 100% noise level.

*4.4.1 Impact of the shared hyper-relational fact encoder.* We observe that NYLON significantly outperforms NYLON (noSE) by 2.58% and 6.93% (on average over datasets) on link prediction and error detection tasks, respectively. This verifies our design choice of using the shared hyper-relational fact encoder, which indeed benefits both link prediction and confidence evaluation.

*4.4.2 Impact of the uncertainty sampling.* We observe that NYLON outperforms NYLON (noUS) by 1.67% and 2.14% (on average over datasets) on link prediction and error detection tasks, respectively, which shows the effectiveness of our uncertainty sampling, where we select the facts whose fact-wise confidence is closest to 0.5 (the case that our confidence evaluator is most uncertain of).

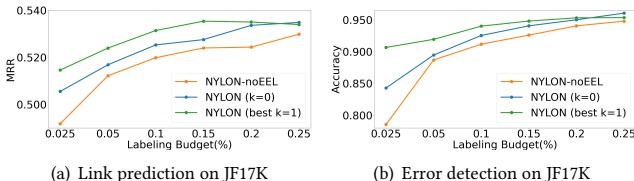

(a) Link prediction on JF17K  (b) Error detection on JF17K

**Figure 3: Tradeoff between performance and labeling budget on JF17K (results on other datasets shown in Appendix A.5)**

*4.4.3 Effort-Efficient Labeling Performance.* We observe that NYLON consistently outperforms NYLON (noEEL), showing the effectiveness of our effort-efficient labeling mechanism, where crowd annotators are only required to label elements until an incorrect one is found. In other words, given the same crowdsourcing budget $b$ (i.e., the number of elements to be checked), our mechanism designed under the "least confidence" principle obtains more but partially labeled facts, while NYLON (noEEL), which requires crowd annotators to label all elements for each query fact, obtains less but fully-labeled facts. NYLON can effectively learn from the larger number of partially labeled facts for confidence evaluation.

Moreover, we show the tradeoff between the performance and the labeling budget in Figure 3. Here we also study the impact of our data augmentation by comparing the number of pseudo-labeled facts $k = 0$ and the *best k* searched over $\{1, 2, 5, 10\}$. First, the performance on both link prediction and error detection tasks increases when increasing the labeling budget, and NYLON achieves a better Pareto frontier than NYLON (noEEL). Furthermore, we see that our data augmentation can effectively boost performance in the case of a small labeling budget. This utility decreases when increasing the labeling budget, and even leads to negative effects for a large budget. Because with enough human-labeled facts for training the confidence evaluator, the pseudo-labeled facts could introduce additional noise. Therefore, $k$ is suggested to be tuned according to the specific labeling budget on each dataset. A detailed parameter sensitivity study of $k$ can be found in Appendix A.6.

## 5 CONCLUSION

In this paper, we study the problem of robust link prediction over noisy hyper-relational facts, and propose NYLON, a Noise-resistant hYper-reLatiONal link prediction technique via active crowd learning. We first introduce the element-wise confidence beyond the traditional fact-wise confidence for hyper-relational facts, and bridge the gap between them using the "least confidence" principle. Following this principle, NYLON systematically integrates a hyper-relational link predictor using the fact-wise confidence for robust prediction, a cross-grained confidence evaluator predicting both element- and fact-wise confidences, and an effort-efficient active labeler selecting informative facts for crowd annotators to label via an efficient labeling mechanism with label-ratio-compliant data augmentation. We evaluate NYLON on three KG datasets against a sizeable collection of baselines. Results show that NYLON achieves superior and robust performance in both link prediction and error detection tasks, and outperforms best baselines by 2.42-10.93% and 3.46-10.65% in the two tasks, respectively.

In the future, we plan to incorporate KG schemas to further assist the labeling process of crowd annotators.

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

# A  APPENDIX

## A.1  NYLON Training Algorithm

Algorithm 1 summarizes the overall training process. After initializing the first set of noisy-labeled training facts (Line 1), we repeat

---

**Algorithm 1:** NYLON Training Algorithm

**Input:** The set of hyper-relational facts $\mathcal{G}$, the window size $w$, the crowdsourcing budget $b$ for each iteration, and the maximum number of iterations $Iter$

**1** Initialize the set of noisy-labeled training facts $\{\mathcal{T}_1\}$ under the crowdsourcing budget $b$;

**2** **for** $i = 1; i \leq Iter; i{+}{+}$ **do**

**3**   Incrementally train the cross-grained confidence evaluator on $\{\mathcal{T}_l \mid \max(i - w, 1) < l \leq i\}$ to minimize $loss_C$ in Eq. 9;

**4**   Train the hyper-relational link predictor on $\mathcal{G}$ to minimize $loss_L$ in Eq. 4, and cache the confidence of all facts;

**5**   Get a new set of noisy-labeled training facts $\{\mathcal{T}_{i+1}\}$ under the budget $b$ using the cached confidence via the effort-efficient active labeler.

**6**   **if** $loss_L$ *has converged* **then**

**7**     break;

---

the three training pipelines: incrementally train the cross-grained confidence evaluator using meta-learning (Line 3), train the hyper-relational link predictor (Line 4) and get a new set of noisy-labeled training facts via the effort-efficient active labeler (Line 5). As our ultimate goal is for robust link prediction over noisy KGs, we use the link prediction loss for the convergence criterion (Line 6). Our code and datasets are available here[4].

## A.2  Baselines and Settings

The three categories of baselines included in our experiments are as follows:

- *Hyper-relational link prediction techniques*, where they do not consider the confidence of noisy facts. **GRAN** [56] represents a hyper-relational fact as a heterogeneous graph and proposes a transformer-based network adopting an edge-biased self-attention mechanism to capture different types of connections in the graph. **StarE** [18] represents a hyper-relational fact as a directed heterogeneous graph and designs a customized message-passing mechanism to extract the inter-vertex interaction using a graph neural network. **Hy-Transformer** [67] extends StarE by replacing the message-passing mechanism with a lightweight entity/relation embedding module and adding a qualifier-oriented auxiliary training task. **QUAD** [47] also extends StarE by adopting two separate aggregators to encode the primary entity-typed triplets and associated key-type pairs, respectively. We do not include NaLP [22], HINGE [41], and NeuInfer [21] here, as they have been shown to underperform the methods above [56].

- *Robust learning techniques* for both error detection and link prediction over noisy KGs. **CKRL** [61] is a robust link prediction technique over noisy KGs. It evaluates the confidence of a fact combining a local triple confidence score computed using an energy function and a global path confidence score computed via the relation path reliability; it is originally proposed for triple facts only and uses TransE [8] as its link predictor. To handle hyper-relational link prediction tasks, we replace TransE with our hyper-relational link predictor and use its loss as the energy score. Moreover, we use the base triplets of hyper-relational facts to compute the global path confidence. **CKRL-Fix** is our improved

---

[4]Github link removed for double-blind review.

version of CKRL [61] by re-designing the local triple confidence to fit the energy score (loss) output by our hyper-relational link predictor. Specifically, CKRL designs a mechanism to reinforce the confidence of facts with high energy scores (greater than a threshold) and punish the confidence of facts with low energy scores (less than the threshold). This threshold is defined as a margin-based distance function, which fits well with the margin-based training strategy of TransE, as pointed out in the paper. However, when replacing TransE with our hyper-relational link predictor, the thresholding mechanism does not fit our cross-entropy loss. Therefore, we propose a ranking-based mechanism to reinforce the confidence of facts whose loss is the top half minimum and punish the confidence of facts whose loss is the bottom half minimum, which can indeed improve the link prediction performance as evidenced by our experiments below. **KGTtm** [31] is an error detection technique for noisy triplets. It integrates confidence scores measured at the entity level using energy scores, at the relationship level via resource allocation, and at the KG global level via reachable paths inference, which are then used to output the probability of an input fact being fake. As KGTtm is originally designed for error detection tasks trained on a fixed dataset without active crowd learning, we thus integrate it with our hyper-relational link predictor by simply averaging the confidence scores at the three levels (without training the binary classifier) as the fact-wise confidence, and combine it with our hyper-relational link predictor using our noisy-resistant link prediction loss. **IDKG** [27] is another error detection technique, which combines resource-allocation-based entity semantic representation confidence scores and relation path confidence scores. Similar to KGTtm, IDKG also uses a classifier to output the probability of an input fact being fake. Subsequently, we adopt the same setting above (as for KGTtm) to integrate IDKG with our hyper-relational link predictor, i.e., combining the averaged confidence scores with our hyper-relational link predictor using our noisy-resistant link prediction loss.

- *Active crowd learning techniques* for selecting informative data samples to label via crowdsourcing.
  - First, following the setting of the error detection techniques [27, 31], the confidence scores can be used as features to learn a binary classifier for predicting the probability of a fact being fake. We can thus regard the output of this classifier as the fact-wise confidence which is then used for active learning and crowdsourcing. In other words, we replace our cross-grained confidence evaluator with this classifier, and keep our link predictor and active labeler. Note that as these classifiers cannot output element-wise confidence, our effort-efficient labeling mechanism is not used. We implement this setting for CKRL-Fix, KGTtm, and IDKG, denoted as **CKRL-Fix (AL)**, **KGTtm (AL)**, and **IDKG (AL)**, respectively; we keep only the improved version of CKRL here.
  - Second, we also consider sample selection techniques for active learning based on learnt embeddings, and integrate these techniques with our NYLON by replacing our uncertainty sampling in Eq 10. Specifically, Farthest-Traversa [48] (FT) selects the facts with the largest embedding distance from the center embedding of all facts. Density-Weighted Methods [44] (DWM) utilizes the fact embedding distance from the center

embedding of all facts to re-scale our fact-level confidence. We use the output of our hyper-relational fact encoder as the fact embedding and compute the center embedding as the average of embeddings of all facts. Note that these sample selection techniques can only be applied to the fact-wise confidence, for a fair comparison, we still use our cross-grained confidence evaluator to predict both element- and fact-wise confidence and apply these baseline techniques to our predicted fact-wise confidence. Meanwhile, our effort-efficient active labeler remains fully functional. As these two techniques re-use most of the components of our proposed NYLON, we thus denote them as **NYLON-FT** and **NYLON-DWM**, respectively. We also refer to them together with our NYLON as the *NYLON family* in our experiments.

In our experiments, NYLON is set with the number of self-attention layers in the shared hyper-relational fact encoder $L_H = 12$ with attention head 4 and embedding size 256, the number of linear layers in our cross-grained confidence evaluator $L_C = 4$, the window size for incremental learning $w = 10$, and the maximum number of iterations $Iter = 100$. We search the best value for the number of pseudo-labeled facts generated for data augmentation $k$ within $\{0, 1, 2, 5, 10\}$.

### A.3 Performance with 40% noise level

Following the setting in the [61], we also show the results with the noise level of 40% on JF17K, WikiPeople, and WD50K. Table 6 and 7 show the link prediction performance and error detection performance, respectively. The results are similar to the case of 100% noise level. Specifically, we observe that our NYLON outperforms the best-performing baselines by 2.12%, 3.38%, and 4.33% on link prediction performance, and by 9.36%, 10.34%, and 13.94% on fact-wise error detection performance, over JF17K, WikiPeople and WD50K, respectively.

### A.4 Performance across different noise levels on WikiPeople and WD50K

Figure 4 shows the performance comparison with the best-performing baselines (per category of techniques) across different noise levels on WikiPeople and WD50K. Similar to the results on JF17K, we see that NYLON is significantly more robust than baseline techniques against different levels of noise. When increasing the level of noise, the performance of NYLON decreases much slower than baseline techniques or even retains its performance sometimes.

### A.5 Tradeoff between model performance and labeling budget on WikiPeople and WD50K

Figure 5 shows the tradeoff between model performance and labeling budget on WikiPeople and WD50K. Similar to the results on JF17K, we see that NYLON consistently outperforms NYLON (noEEL) by achieving a better Pareto frontier. Moreover, the improvement of NYLON (best k) over NYLON (k=0) also decreases when increasing the labeling budgets; sometimes NYLON (best k) even underperforms NYLON (k=0) for a large labeling budget. To further study this issue, we present below a parameter sensitivity study of the number of pseudo-labeled facts $k$.

**Table 6: Overall link prediction performance with 40% noise level**

| Method | | JF17K | | | | WikiPeople | | | | WD50K | | | |
|---|---|---|---|---|---|---|---|---|---|---|---|---|---|
| | | Entity | | Relation | | Entity | | Relation | | Entity | | Relation | |
| | | MRR | Hit@1 | MRR | Hit@1 | MRR | Hit@1 | MRR | Hit@1 | MRR | Hit@1 | MRR | Hit@1 |
| Hyper-relational link prediction | GRAN | 0.5079 | 0.4235 | 0.9916 | 0.9876 | 0.4563 | 0.3634 | 0.9621 | 0.9389 | 0.3062 | 0.2448 | 0.9190 | 0.8842 |
| | StarE* | 0.4222 | 0.3312 | N/A | N/A | 0.3496 | 0.2184 | N/A | N/A | 0.2531 | 0.1792 | N/A | N/A |
| | Hy-Transformer* | 0.4854 | 0.3941 | N/A | N/A | 0.3904 | 0.2667 | N/A | N/A | 0.2800 | 0.2071 | N/A | N/A |
| | QUAD* | 0.3959 | 0.2930 | N/A | N/A | 0.3280 | 0.1987 | N/A | N/A | 0.2435 | 0.1724 | N/A | N/A |
| Robust learning | CKRL | 0.4872 | 0.4022 | 0.9907 | 0.9860 | 0.4661 | 0.3712 | 0.9535 | 0.9304 | 0.2982 | 0.2361 | 0.8963 | 0.8569 |
| | CKRL-Fix | 0.5075 | 0.4250 | 0.9891 | 0.9834 | 0.4639 | 0.3745 | 0.9531 | 0.9306 | 0.3068 | 0.2452 | 0.9012 | 0.8640 |
| | KGTtm | 0.4784 | 0.3923 | 0.9833 | 0.9756 | 0.4493 | 0.3559 | 0.9516 | 0.9252 | 0.2966 | 0.2362 | 0.8932 | 0.8539 |
| | IDKG | 0.5069 | 0.4247 | 0.9900 | 0.9848 | 0.4786 | 0.3906 | 0.9558 | 0.9340 | 0.3041 | 0.2430 | 0.9021 | 0.8655 |
| Active crowd learning | CKRL-Fix (AL) | 0.5145 | 0.4296 | 0.9895 | 0.9843 | 0.4667 | 0.3770 | 0.9550 | 0.9335 | 0.3105 | 0.2486 | 0.9022 | 0.8658 |
| | KGTtm (AL) | 0.5076 | 0.4227 | 0.9892 | 0.9833 | 0.4577 | 0.3658 | 0.9522 | 0.9295 | 0.3072 | 0.2460 | 0.9003 | 0.8635 |
| | IDKG (AL) | 0.5079 | 0.4253 | 0.9888 | 0.9833 | 0.4606 | 0.3712 | 0.9538 | 0.9317 | 0.3063 | 0.2451 | 0.9001 | 0.8632 |
| NYLON family | NYLON-FT | 0.5242 | 0.4377 | 0.9919 | 0.9882 | 0.4945 | 0.4066 | 0.9594 | 0.9418 | 0.3241 | 0.2608 | 0.9119 | 0.8796 |
| | NYLON-DWM | 0.5205 | 0.4349 | 0.9918 | 0.9879 | 0.4926 | 0.4018 | 0.9576 | 0.9377 | 0.3235 | 0.2601 | 0.9109 | 0.8769 |
| | **NYLON** | **0.5349** | **0.4482** | **0.9924** | **0.9888** | **0.5035** | **0.4169** | **0.9667** | **0.9493** | **0.3348** | **0.2707** | **0.9201** | **0.8886** |

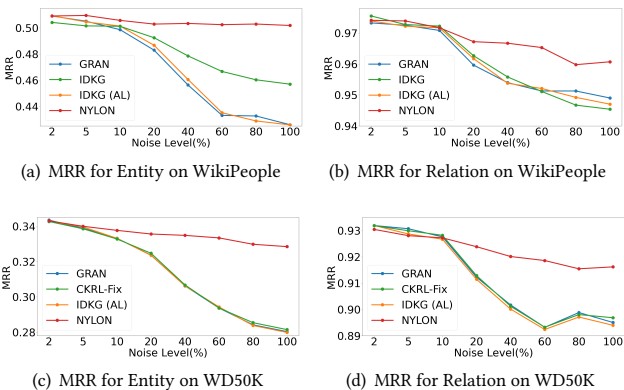

(a) MRR for Entity on WikiPeople    (b) MRR for Relation on WikiPeople

(c) MRR for Entity on WD50K    (d) MRR for Relation on WD50K

**Figure 4: Performance comparison with the best-performing baselines (per category of techniques) on WikiPeople and WD50K**

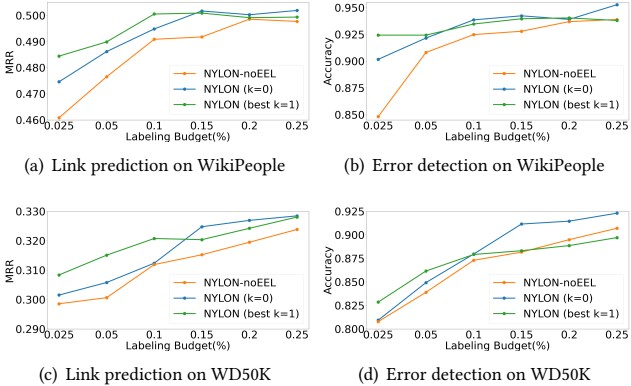

(a) Link prediction on WikiPeople    (b) Error detection on WikiPeople

(c) Link prediction on WD50K    (d) Error detection on WD50K

**Figure 5: Tradeoff between model performance and labeling budget on WikiPeople and WD50K**

**Table 7: Error detection performance (in accuracy) with 40% noise level**

| Method | JF17K | | WikiPeople | | WD50K | |
|---|---|---|---|---|---|---|
| | Fact | Element | Fact | Element | Fact | Element |
| CKRL-Fix (AL) | 0.4957 | N/A | 0.4995 | N/A | 0.4999 | N/A |
| KGTtm (AL) | 0.5001 | N/A | 0.6876 | N/A | 0.5801 | N/A |
| IDKG (AL) | 0.6890 | N/A | 0.7512 | N/A | 0.6597 | N/A |
| NYLON-FT | 0.8784 | 0.9472 | 0.8635 | 0.9281 | 0.8102 | 0.8965 |
| NYLON-DWM | 0.8325 | 0.9292 | 0.7754 | 0.8821 | 0.7577 | 0.8747 |
| NYLON | **0.9606** | **0.9791** | **0.9528** | **0.9661** | **0.9231** | **0.9543** |

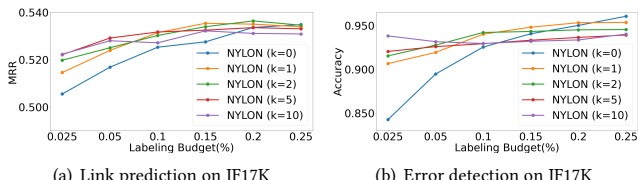

(a) Link prediction on JF17K    (b) Error detection on JF17K

**Figure 6: Impact of augmentation amount**

## A.6 Parameter sensitivity study of the number of pseudo-labeled facts $k$

We study the impact of the number of pseudo-labeled facts generated for data augmentation $k \in \{0, 1, 2, 5, 10\}$. Figure 6 shows the results on JF17K. We observe that in the case of a very small labeling budget (e.g., 0.025%), a larger $k$ yields better performance in general. Because given very limited human-labeled facts, the pseudo-labeled facts can effectively help train the confidence evaluator. However, this utility decreases when increasing the labeling budget. In the case of a large labeling budget (e.g., 0.25%), a larger $k$ yields even worse performance. Because when the amount of human-labeled facts is enough to train the confidence evaluator, the pseudo-labeled facts could introduce additional noise. Therefore, $k$ is suggested to be tuned according to the specific labeling budget on each dataset.

Received 20 February 2007; revised 12 March 2009; accepted 5 June 2009