# OpenReview forum: "Robust Link Prediction over Noisy Hyper-Relational Knowledge Graphs"
_ACM.org/TheWebConf/2024/Conference — TheWebConf24 Oral_

### Official Review · Reviewer_hebp · 2023-10-31

**Novelty:** 4
**Technical Quality:** 4

**Review:**

This work proposes NYLON, an approach to tackle knowledge graph completion for hyper-relational KGs, with a focus on robustness and handling noisy data. Experiments on three datasets demonstrate that the proposed approach outperforms baselines.

- I wonder if the confidence scores predicted by the confidence evaluator are well-calibrated or not. To me, a well-calibrated confidence mechanism would greatly help resolve noisy KG instances.

- The confidence evaluator seems to be based on linear layers and a concatenation of entity/relation vectors. However, evaluating the confidence/correctness of a factual association (represented as a KG instance) requires priors/relevant knowledge. Why do the authors believe, that without access to external/relevant information, a linear layer is capable of judging the confidence/factuality of facts? An alternative would be to employ a language model or retrieval system as they could both be sources of knowledge and information.

- In Table 2, the validity of the relation prediction task for JF17K is questionable: all models perform .98+ on MRR and Hit @1. In this case, the improvement performance on a magnitude of 1e-3 is really just overfitting, not indicative of any real progress.

- It would be great to provide other metrics, in addition to the strict ones (MRR and Hit @1) such as Hit @3/5/10. This would better establish NYLON as a robust approach.

- Certain ablation studies indicate that the contribution of different components could be marginal, but this isn't necessarily a serious concern though.

- It would be great to include a qualitative analysis: specific KG instances, predictions, confidence scores, etc. to provide a better sense of how NYLON works and why it is better in practice.

**Questions:**

please see above

**Reviewer Confidence:**

3: The reviewer is confident but not certain that the evaluation is correct

**Scope:**

3: The work is somewhat relevant to the Web and to the track, and is of narrow interest to a sub-community

---

### Official Review · Reviewer_jZhx · 2023-11-12

**Novelty:** 6
**Technical Quality:** 6

**Review:**

Summary:
This paper presents a noisy hyper-relational KG link prediction method, NYLON, that uses active learning. NYLON incorporates element-wise and fact-wise confidence assessments, connected through a "least confidence" principle for efficient crowd labeling. It integrates a hyper-relational link predictor, a confidence evaluator, and an active labeler to improve labeling efficiency and data augmentation. Tested on KG datasets, NYLON demonstrates superior performance in link prediction and error detection, outperforming baseline methods.


Strength:
1. Strong experimental performance, outperforming all baseline methods
2. Proposed a new setting for noisy hyper-relational KG link prediction.


Weakness:
1. The paper is not in review format, which has line numbers on the page edge.
2. I recommend that the authors rephrase the title of this paper add some active learning related words to help it easily identify by other researchers.
3. As the authors claim that NYLON has an effort-efficient labeling mechanism, it would be better to present the labeling effort comparison in the paper.
4. In the ablation study, the performance under the no-EEL setting seems very close to the full model, especially in the link prediction task. I would be happy to see authors discussing this phenomenon.
5. What is the model architecture specifically for the error detection task? It seems the architecture presented in the methodology section can only be applied to link prediction task.
6. How the NYLON model works in the inference stage is unclear.
7. Since the active learning process needs human involvement, while human have their personal bias in labeling. It would be interesting to see the robustness of NYLON when different users are involved in the active learning process and its robustness for personal bias.
8. The confidence score prediction is an important module in the proposed NYLON framework. Its mechanism and effectiveness need more experiments to validate. I would encourage authors to do a case study to present the confidence score for some correct and incorrect facts to demonstrate the effectiveness of NYLON’s confidence module.

**Questions:**

Please see the review above.

**Reviewer Confidence:**

3: The reviewer is confident but not certain that the evaluation is correct

**Scope:**

3: The work is somewhat relevant to the Web and to the track, and is of narrow interest to a sub-community

---

### Official Review · Reviewer_z1q8 · 2023-11-22

**Novelty:** 6
**Technical Quality:** 6

**Review:**

The paper addresses the problem of link prediction over noisy Knowledge Graphs (KG) and highlights the challenges of confidence assessment in hyper-relational facts. Hyper-relational KGs not only have triple facts but also have numerous key-value pairs to provide further information about the triple fact. Noise in such KG is inevitable and significantly affects link prediction. Moreover, existing techniques for link prediction of hyper-relational KGs have certain limitations.

The authors propose NYLON, a noise-resistant link prediction technique using active crowd learning. This methodology utilizes element-wise confidence to measure the confidence of each entity and relation, combined with fact-wise confidence via the “least confidence” principle. Once the least confident fact is selected, the crowd annotators check the elements of the fact according to the ascending order of its element-wise confidence and terminate labeling until one incorrect element is found. Such an effort-efficient labeling mechanism helps in selecting informative and reducing the number of facts for annotators to label.

# Strengths
The authors gave a good overview of the current baseline in link prediction over KGs.
The proposed novel methodology is robust and well-described. NYLON outperforms existing baselines both in link prediction and error detection tasks.
The findings of the paper can help to improve the reliability of link prediction over noisy KGs, thus enhancing the performance of further analysis performed on KGs.

# Weakness
Details of the annotation process and the number of annotators involved in effort-efficient labeling are not explicitly mentioned.
In this approach, the performance of the model heavily relies on the quality of annotations provided by the crowd. If the annotators are not well-informed or biased this may affect the overall performance of the learning model.

# Errors
None

**Questions:**

How do authors deal with poor quality annotations?
How many human-labeled facts would be enough to train a confidence evaluator?

**Ethics Review Description:**

-

**Reviewer Confidence:**

4: The reviewer is certain that the evaluation is correct and very familiar with the relevant literature

**Scope:**

4: The work is relevant to the Web and to the track, and is of broad interest to the community

---

### Official Review · Reviewer_1sjo · 2023-11-23

**Novelty:** 6
**Technical Quality:** 5

**Review:**

This paper presents the NYLON (Noise-resistant hYper-reLatiONal link prediction) method for improving predictions in noisy hyper-relational Knowledge Graphs. Hyper-relational facts here are defined by a triplet linked to multiple key-value pairs. NYLON's performance was superior to several baseline models, as demonstrated in three real-world KG datasets.

The paper is well-composed and clear, addressing a pertinent topic. The challenge of hyper-relational facts, significant in various practical scenarios, is intriguingly tackled in the paper. While the solution proposed is not highly innovative, it is effective across three established benchmarks.

The evaluation is comprehensive, testing the approach against 11 baselines on three knowledge graphs. Both the error analysis and ablation study provide valuable insights. It is recommended that the evaluation includes statistical testing, as some improvements appear minimal and may not be statistically significant. A significant shortcoming of the paper is the lack of publicly available code or data. The availability of these resources is a standard expectation in contemporary academic research.

**Questions:**

Can you share the material on an anonymous link?

**Ethics Review Description:**

no concerns

**Reviewer Confidence:**

3: The reviewer is confident but not certain that the evaluation is correct

**Scope:**

4: The work is relevant to the Web and to the track, and is of broad interest to the community

---

### Official Review · Reviewer_2rF1 · 2023-11-24

**Novelty:** 4
**Technical Quality:** 3

**Review:**

The paper presents a robust link prediction method called NYLON for hyper-relational KGs. NYLON incorporates fact-wise confidence and element-wise confidence, enabling efficient crowd labeling using a least confidence principle. Experimental results on three datasets show that NYLON achieves state-of-the-art performance in both link prediction and error detection tasks on noisy KGs. However, there are some weaknesses that need to be addressed:
1. It is important to mention that the core module of NYLON, the hyper-relational link predictor, is actually GRAN[1]. This information is missing in the paper.
2. In the experiments, the baselines for hyper-relational link prediction are not the most up-to-date. There have been recent advancements in this area, such as HAHE[2], HyperFormer[3], ShrinkE[4], HyConvE[5], TransEQ[6], HAET[7], and LGHAE[8], which should be included for comparison.
3. The extensive utilization of large language models has yielded remarkable achievements in various domains. How about the capabilities of these models in addressing robust link prediction tasks on noisy hyper-relational knowledge graphs.

[1] Wang Q, Wang H, Lyu Y, et al. Link prediction on n-ary relational facts: A graph-based approach[C]. ACL, 2023.

[2] Luo H, Yang Y, Guo Y, et al. HAHE: Hierarchical Attention for Hyper-Relational Knowledge Graphs in Global and Local Level[C]. ACL, 2023.

[3] Hu Z, Gutiérrez-Basulto V, Xiang Z, et al. HyperFormer: Enhancing entity and relation interaction for hyper-relational knowledge graph completion[C]. CIKM, 2023.

[4] Xiong B, Nayyer M, Pan S, et al. Shrinking Embeddings for Hyper-Relational Knowledge Graphs[C]. ACL, 2023.

[5] Wang C, Wang X, Li Z, et al. HyConvE: A Novel Embedding Model for Knowledge Hypergraph Link Prediction with Convolutional Neural Networks[C]. WWW. 2023.

[6] Liu Y, Yang S, Ding J, et al. Two Birds, One Stone: An Equivalent Transformation for Hyper-relational Knowledge Graph Modeling[J]. ICLR, 2023.

[7] Ma T, Huang L, Xue H. Improving Hyper-relational Knowledge Graph Representation with Multi-grained Encoding[C]. DASFAA, 2023.

[8] Yuan P, Qi Z, Sun H, et al. LGHAE: Local and Global Hyper-relation Aggregation Embedding for Link Prediction[C]. ICAPSS, 2023.

**Questions:**

No

**Reviewer Confidence:**

4: The reviewer is certain that the evaluation is correct and very familiar with the relevant literature

**Scope:**

4: The work is relevant to the Web and to the track, and is of broad interest to the community

---

### Decision · Program_Chairs · 2024-01-22

**Decision:**

Accept (Oral)

**Comment:**

The reviews collectively recognize the paper's strength in presenting NYLON, a robust link prediction method for hyper-relational KGs, exhibiting superior performance over baselines. However, common concerns across reviews include the omission of NYLON's core module origin (GRAN), outdated baselines, the absence of publicly available code and data, and the need for statistical testing. Additionally, addressing the impact of human annotator quality and bias, clarifying aspects like model architecture for error detection and the inference stage, and providing a more comprehensive evaluation with additional metrics are highlighted as essential improvements.

 The authors have addressed the identified weaknesses in their rebuttal in a satisfactory manner and proposed ways to address those in the CR.

 In the light of the above, I suggest accepting this paper.